# Retaining Knowledge for Learning with Dynamic Definition

**Zichang Liu**
Rice University
Houston, TX 77025
zichangliu@rice.edu

**Benjamin Coleman**
Rice University
Houston, TX 77025
brc7@rice.edu

**Tianyi Zhang**
Rice University
Houston, TX 77025
tz21@rice.edu

**Anshumali Shrivastava**
Rice University/ThirdAI Corp.
Houston, TX 77025
anshumali@rice.edu

## Abstract

Machine learning models are often deployed in settings where they must be constantly updated in response to the changes in class definitions while retaining high accuracy on previously learned definitions. A classical use case is fraud detection, where new fraud schemes come one after another. While such an update can be accomplished by re-training on the complete data, the process is inefficient and prevents real-time and on-device learning. On the other hand, efficient methods that incrementally learn from new data often result in the forgetting of previously-learned knowledge. We define this problem as Learning with Dynamic Definition (LDD) and demonstrate that popular models, such as the Vision Transformer and Roberta, exhibit substantial forgetting of past definitions. We present a first practical and provable solution to LDD. Our proposal is a hash-based sparsity model *RIDDLE* that solves evolving definitions by associating samples only to relevant parameters. We prove that our model is a universal function approximator and theoretically bounds the knowledge lost during the update process. On practical tasks with evolving class definition in vision and natural language processing, *RIDDLE* outperforms baselines by up to 30% on the original dataset while providing competitive accuracy on the update dataset.

## 1 Introduction

**Motivation:** A common machine learning pipeline is to first define a task objective based on historical experience, then deploy a model trained to optimize the objective over collected data. However, machine learning models rarely operate in such a static environment. Instead, models must be dynamically updated to accommodate changes in class definitions without forgetting previous knowledge. Consider the case of fraud detection, where models must adapt to emerging scam patterns while also providing protection against previously-known patterns. Machine learning models are also used to identify harmful microbes given gene sequences, where it is important to perform well on new variants or newly discovered microorganisms. A similar situation arises when models are used to filter inappropriate content such as hate speech and violence. New phrases or previously-innocuous language may become inappropriate due to world events or attempts by users to circumvent the filter. While the classification task remains the same (fraud or safe, harmful or not, inappropriate or appropriate, etc.), class definitions evolve over time and likely differ significantly from the point of the initial training. We refer to this setting as *Learning under Dynamic Definition* (LDD). It should be

36th Conference on Neural Information Processing Systems (NeurIPS 2022).

noted that this notion is very different for several other notions of distribution shift in the literature, which we review in the next section.

A standard solution is to retrain the model on all available data (original dataset and data for the new definitions). This methodology is inefficient due to data storage and extra computation, and even prohibitive when dealing with large models and streaming data. Furthermore, it prevents real-time and on-device learning. An ideal and efficient solution is to update the trained model incrementally only with data for the new definitions (e.g. using online gradient descent). However, this is a challenging process due to the tendency of machine learning models, in particular neural networks, to forget previous knowledge. This situation, known as collapse/catastrophic forgetting [25], causes performance decay on the old definitions.

**Our Proposal:** Catastrophic forgetting occurs because gradient updates from training examples in the update process are able to change parameters that are tuned for optimal performance on the original dataset [30]. Inspired by recent connections between random partitions and distribution models [19], we propose RIDDLE (Retaining Information in a Dynamically Defined Learning Environment), a novel architecture with meaningful parameter

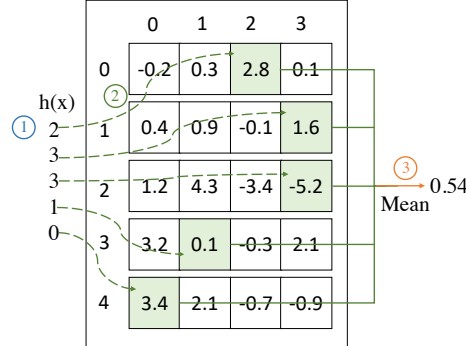

Figure 1: The figure demonstrates a toy RIDDLE model with $L = 5$ and $R = 4$. Given an input $x$, we calculate its hash code $h(x)$ and access the weights at those locations. We return the average as the output.

groupings. RIDDLE works by partitioning the input space and assigning a set of parameters to each partition. Gradient updates are restricted only to locally relevant parameters, identified by cheap hashing computations. RIDDLE can be viewed as a Mixture-of-Expert-Layer where the gating function is Locality Sensitive Function (LSH) and each expert is just one number. At the same time, RIDDLE can be viewed as a kernel method where the prediction is an efficient weighted kernel density estimation. RIDDLE can be plugged into diffrent neural network architectures, just like fully connect layers, to ease the memory loss while utilizing the learned features, as shown in Figure 2. We summarize our technical contributions below.

- We introduce the RIDDLE (in Section 3), consisting of a set of parameters indexed by locality-sensitive hash functions. RIDDLE is computationally efficient and requires less memory than popular machine learning models with similar accuracy.

- We prove that RIDDLE is universal function approximator and can therefore represent any function( Section 4). We prove a formal connection between parameter usage and distribution similarity in Theorem 3.2.

- In Section 5, we apply RIDDLE in the dynamic definition setting for four real-world tasks. Our method outperforms baselines by up to 30% on the original dataset while achieving competitive accuracy on the new dataset.

## 2 Related Work

**Continual Learning:** Continual learning refers to learning and remembering multiple tasks from a stream of information. Current approaches can be split into three main categories (1) memory replay to complement neural networks [45, 29], (2) constraints on network updates [38, 28, 52], and (3) dynamic architectural adjustments to accommodate new information, such as additional layers [49, 61]. Recently, few-shot learning methods are also applied into continue learning to solve sequential learning with limited data [8, 58, 53]. Continual learning methods encounter similar challenges to the ones presented by learning under dynamic distribution, most notably catastrophic forgetting. However, continual learning attempts to solve a sequence of related tasks, often expressed as task embeddings, over data that does not necessarily reflect changes in class definitions.

**Train-Test Distribution Shift:** Transfer learning and test-time adaptation attempt to remedy misalignments between the training and test distribution. Although such methods may seem relevant in learning with dynamic definition, they have dramatically different goals. Transfer learning attempts to

adapt a model from a data-rich source domain to a data-scarce target domain, without regard for the performance on the source domain [65, 56, 59]. While several works on train-test distribution shift focus on robustness, it is from the perspective of out-of-distribution detection [24, 39] or uncertainty estimation [44, 33, 57]. Our objective is to provide models that perform well on both the original dataset and new dataset, where class definitions are different, rather than simply detect or adapt to the distribution change.

**Online Learning:** Online learning regards optimization as a process where a model is trained by interacting with a stream of data. Online learning algorithms attempt to minimize the *regret*, or difference between the loss of the predictions by the incrementally-trained model and the loss of the globally optimal model [48, 46]. Online learning methods attain minimal regret against adaptive adversaries and could be expected to perform well in the dynamic distribution setting. However, the best performing methods for non-convex online learning require retraining over all previous data at every time step [55], making them practically unsuitable for the tasks we consider.

**Locality-Sensitive Hashing in Machine Learning:** Locality-sensitive hash (LSH) functions are a central component of our proposal. An LSH family is a set of functions that maps similar inputs to the same hash value (with high probability). LSH is an incredibly powerful and well-studied framework, with the following formal definition [27, 7, 20, 4].

**Definition 2.1 ($(S_0, cS_0, p_1, p_2)$-sensitive hash family).** A family $\mathcal{H}$ is called $(S_0, cS_0, p_1, p_2)$-sensitive with respect to a similarity function $sim(\cdot, \cdot)$ if for any two points $x, y \in \mathbb{R}^D$ and $h$ chosen uniformly from $\mathcal{H}$ satisfies:

- If $sim(x, y) \geq S_0$ then $Pr[h(x) = h(y)] \geq p_1$

- If $sim(x, y) \leq cS_0$ then $Pr[h(x) = h(y)] \leq p_2$

The probability $Pr[h(x) = h(y)]$ is known as the collision probability of $x$ and $y$, where we use the term "collision" to mean that two points map to the same hash value. For the purpose of our arguments, we use a stronger notion of LSH, in which the collision probability is a monotonic increasing function of the similarity between $x$ and $y$. That is $Pr[h(x) = h(y)] \propto f(\mathrm{sim}(x, y))$.

Under the above conditions, the collision probability $Pr[h(x) = h(y)]$ forms a positive semi-definite radial kernel [19]. A number of useful kernels can be obtained via LSH, including MinHash for the Jaccard kernel [12], signed random projections for the angular kernel [15], and p-stable LSH for a kernel in Euclidean space [20].

Recent works have shown that LSH can be used for efficient statistical estimation [54, 14, 41, 19, 35, 36, 37, 62], compressive machine learning [34], nearest neighbor search [5, 7, 6] and to accelerate neural network training and inference [17, 16, 32, 40, 63]. A particularly relevant result is the RACE sketch [19], which solves kernel density estimation using the number of examples that fall into the random partitions induced by an LSH function.

## 3 RIDDLE for Learning with Dynamic Definition

**Learning with Dynamic Definition:** We formally define the problem as follows.

**Definition 3.1.** Given a model $f(x; \theta)$ and a loss $\ell(f(x; \theta), y)$, consider an original dataset $D^o = \{(x_i^o, y_i^o)\}_{i=1}^m$ drawn i.i.d. from a distribution $P^o$ and an update dataset $D^u = \{(x_i^u, y_i^u)\}_{i=0}^n$ drawn i.i.d from distribution $P^u$. Let $\theta^o$ be the parameters that optimize $\mathcal{L}_o(\theta) = \sum_{D^o} \ell(f(x; \theta), y)$. The dynamic definition learning problem is to find a solution minimizing $\mathcal{L}_{o+u}(\theta)$ when we are only given access to $\theta^o$ and $D^u$.

$$\mathcal{L}_{o+u}(\theta) = \sum_{(x,y) \in D^o} \ell(f(x; \theta), y) + \sum_{(x,y) \in D^u} \ell(f(x; \theta), y)$$

In this paper, we consider the model performance in response to the changes from $P^o$ to $P^u$, specifically by allowing $x$ to be drawn from different input spaces. Note that Definition 3.1 generalizes in a straightforward way to sequences of dataset updates by considering $D^o$ to be the union of all prior data. The difficulty of the problem may be expressed in terms of the Kullback-Leibler divergence $D_{KL}(P^o \| P^u)$. Intuitively, small changes in distribution are easier to handle because $\theta$ need not change substantially to perform well on $D^u$.

## 3.1 Architecture

RIDDLE is a 2D parameter array $S$ that contains $L$ rows and $R$ columns $S \in \mathbb{R}^{L \times R}$. The model is indexed by a set of $L$ randomly generated LSH functions $h_l \to \{1, \dots R\} \in \mathbb{Z}$ for $l = 1, \dots L$. The prediction evaluated at a query $x$ is defined as following:

$$f(x; S, h) = \frac{1}{L} \sum_{l=1}^{L} S[l, h_l(x)]$$

Algorithmically, the process consists of three steps: (1) hash code computation, (2) partition look-ups, and (3) output averaging. Specifically, the model output for a query $x$ is an average over the $L$ parameters retrieved from $S$ using the hash codes $h(x)$. Figure 1 illustrates a toy example.

For a classification problem with $C$ classes, the 2D array is repeated $C$ times, resulting in a 3D tensor of parameters $S \in \mathbb{R}^{C \times L \times R}$. The same LSH functions are used for all $C$ repetitions, and the classification prediction for query $x$ is defined as follows:

$$f(x; S, h) = \text{argmax}_{c \in C} \left( \frac{1}{L} \sum_{l=1}^{L} S[c, l, h_l(x)] \right)$$

RIDDLE can be used as a stand alone machine learning model on simple datasets but can also be combined with existing neural networks. Specifically, RIDDLE can use the learned representation from neural network (Figure 2). RIDDLE can also be implemented in standard deep learning frameworks and trained using SGD (Algorithm 1). It should be noted that the parameters of LSH functions $h(x)$ are not trainable. Only the values in $S \in \mathbb{R}^{L \times R}$ will be updated by SGD. The inference process consists only of inexpensive hash evaluations and memory look-ups. We use LSH functions based on sparse projections [2], which compute $h(x)$ via an inner product with parameter vectors randomly drawn from $\{-1, 0, 1\}$ with probability $\{1/6, 2/3, 1/6\}$. With such a hash function, computation can be reduced to addition, subtraction and memory lookup.

## 3.2 Intuition

RIDDLE may seem conceptually different from neural networks at first gasp, however, RIDDLE can be viewed as a combination of Mixture-of-Expert-Layers [23] with Kernel Methods [26]. To understand RIDDLE as a mixture of experts, observe that the gating function for every row in RIDDLE is a randomly initialized hashing function $h_i$, and the expert is one number stored at $S[i, h_i(x)]$. RIDDLE can be viewed as an ensemble of $L$ such Mixture-of-Expert-Layers. On the other side, RIDDLE is also a kernel density estimation process where the LSH collision probability is the kernel function. The number stored at $S[i, h_i(x)]$ is an efficient estimation of the weighted kernel values between the test sample and training samples.

## 3.3 RIDDLE for Learning with Dynamic Definition

Before we determine the class of functions that are representable using our model, we first demonstrate that this architecture addresses the problem from Definition 3.1. We consider a RIDDLE model $f(x; S, h)$ following the notation from Section 3.1. That is, we are given a model with parameters $\theta = (S, h)$ trained on $D^o$. We wish to adapt the model on $D^u$ using gradient descent.

**Parameter Separation:** The RIDDLE partitions parameters in such a way that inputs from different distributions are likely to use different parameters. The parameter update may be written as:

$$S_{t+1} = S_t[l, h_l(x)] + \eta g[l] \text{ for } l = 1 \dots L, \text{ where } g = \nabla \ell \left( \frac{1}{L} \sum_{l=1}^{L} S[l, h_l(x)], y \right)$$

From this expression, we can see that $L$ parameters are involved in the computation of $f(x; S, h)$ - specifically, the parameters selected by the hash functions $\{h_1(x)...h_L(x)\}$. Based on the LSH property (Definition 2.1), we expect unrelated inputs to have different hash values and therefore use different parameters. When $P^u$ is significantly different from $P^o$, $x^u \in D^u$ and $x^o \in D^u$ are likely to activate different partition and thus use different parameter sets.

---

**Algorithm 1** RIDDLE Training

---

**Input:** Training Dataset $D$, $|D| = m$, number of rows $L$, number of cells $R$, learning rate $\eta$, error function $E(\cdot)$, number of epochs $e$, batch size $b$, random seed $s$.
**Output:** Trained Model $S$, Counters $C$
**Initialize:** Zero initialize $S \in \mathbb{R}^{L \times R}$; Zero initialize counters $C \in \mathbb{R}^{L \times R}$; Randomly generated $L$ independent LSH functions $h_1, \ldots, h_L$ using random seed $s$;
**for** $i = 1 \rightarrow e$ **do**
    Shuffle dataset $D$
    **for** $j = 0 \rightarrow m/b - 1$ **do**
        Take $b$ samples $\{x_{jb}, x_{jb+1}, \ldots, x_{(j+1)b-1}\}$ with labels $\{y_{jb}, y_{jb+1}, \ldots, y_{(j+1)b-1}\}$
        Compute gradient $g = \frac{1}{m} \nabla_S \sum_i E(f_{S,h}(x_i), y_i)$
        Apply update $S \rightarrow S - \eta g$
    **end for**
**end for**
**for** every $x_i \in D$ **do**
    Increase counters $C[l, h_l(x_i)] += 1$ *for* $l = 1 \ldots L$
**end for**

---

We formalize this intuition below by showing that the difference between $f(x; S, h)$ and its initialization is bounded by an estimate of the distribution $P^o$.

**Theorem 3.2.** *Given a dataset $D$, train a RIDDLE model $f(x; S, h)$ by running Algorithm 1 for $e$ epochs with learning rate $\eta$ and initialization $S_0$. Suppose the gradient norms are bounded by $G$. Then*

$$|\mathbb{E}_h[f(x; S, h) - f(x; S_0, h)]| \leq e\eta G \, \mathrm{KDE}(x, D)$$

*where $\mathrm{KDE}(x, D)$ is the kernel density of $x$ over $D$ using the kernel of the LSH function $h$.*

Theorem 3.2 allows us to prove the following guarantee:

**Theorem 3.3.** *Using the notation from Definition 3.1, let $S_o$ be the sketch that optimizes the loss $\mathcal{L}_o(S)$ over $D^o$ and $S_{o+u}$ be the sketch obtained by training over $D^u$ for $e$ epochs with learning rate $\eta$ and initialization $S_o$. If the loss is L-Lipschitz with G-bounded gradients, then:*

$$\mathbb{E}_h[\mathcal{L}_o(S_{o+u}) - \mathcal{L}_o(S_o)] \leq \sum_{x \in D^o} GLe\eta \mathrm{KDE}(x, D^u)$$

The loss difference in Theorem 3.3 quantifies the amount of information about $D^o$ that is forgotten by the model when trained on $D^u$. The excess loss on example $x \in D^o$ is bounded by the kernel density estimate $\mathrm{KDE}(x, D^u)$. When $D^o$ and $D^u$ are drawn from different distributions, which is the characteristic of learning under dynamic distribution, the density becomes small (i.e. as the divergence $D_{KL}(P^o \| P^u) \rightarrow \infty$, $\mathrm{KDE}(x, D^u) \rightarrow 0$).

**Parameter Importance:** The RIDDLE allows us to determine the importance level of each parameter for encoding information from the original distribution with simple counting. We may do this by counting the number of inputs from $D^o$ that fall into each partition, using a set of counters $C \in \mathbb{R}^{L \times R}$. Parameters with a large counter $c$ can be regarded as important for the original distribution, as they participate in prediction for a large fraction of examples from $D^o$. Our intuition is that parameters corresponding to partitions with large count values are critical for performance on $D^o$ and should not be adjusted. We use this idea to improve performance via partition-dependent learning rates. Specifically, we allow the learning rate $\eta(c)$ depend on the partition count $c$. We find that the following update rule provides good empirical results:

$$\eta(c) \propto \frac{\bar{c} + \alpha}{c \log c + 1}, \text{ where } \alpha \text{ is a constant}$$

## 4 RIDDLE is a Universal Function Approximator

In this section, we analyze the class of functions that can be represented using the RIDDLE. Using techniques from the sketching literature, we prove a probabilistic guarantee.

**Intuition:** To see why it is reasonable for our model to be a universal function approximator, observe that the function learned by RIDDLE is a piecewise-constant spline function. Splines are an incredibly powerful representation that can approximate any function, given sufficient flexibility in choosing the partitions (or knots) of the spline [51, 50]. For example, many neural networks can be written as compositions of piecewise-linear splines over a constrained set of partitions[11, 10, 9, 60]. In our case, the partition boundaries are determined by the intersection of LSH functions. With sufficient independent overlapping partitions (large $L$), our model attains a high-quality representation.

**Proof Sketch:** We prove that the expected output of our model is a universal function approximator, where the expectation is taken over the choice of LSH functions. We begin with a proof that any continuous and bounded function can be approximated arbitrarily well by a weighted kernel sum over an $N$ point dataset. We then develop a process that constructs a RIDDLE $S \in \mathbb{R}^{L \times R}$ whose expected output is this kernel sum. Using Chernoff bounds, we show that the error between the model and the kernel sum becomes arbitrarily small as $L \to \infty$ (with high probability). Taken together, we have a probabilistic guarantee for the error between the RIDDLE and any continuous and bounded function.

**Simplified optimization in practice:** Constructing a RIDDLE over a carefully chosen $N$ point dataset can approximate any continuous and bounded function. However, learning both the data and data weights with $N$ being arbitrarily large poses difficulty in optimization. We simplify the optimization by observing that the output of the learning process is not a dataset but a model. Therefore, we consider the easier task of learning the model parameters values directly in practice as shown in Algorithm 1. This is a slight relaxation of the problem, however, we argue that with a row-sum constraint and a reasonable uniqueness assumption about the hash partitions, the two processes are equivalent in practice. Thus, the RIDDLE is a universal function approximator.

## 4.1 Weighted LSH Kernel Sums are Universal Function Approximators

We are interested in the representation capabilities of weighted kernel sums over an $N$-point dataset $D = \{x_1, ...x_N\}$. We consider the specific case where the kernel is the collision probability of an LSH function. Such kernels are referred to in the literature as *LSH kernels* [19].

$$f(x) = \sum_{i=1}^{N} \alpha_i \mathcal{K}(x, x_i)$$

The crucial property is that LSH kernels are *universal*. Informally, a kernel is universal if a weighted kernel sum over a carefully-chosen point set can approximate any function with arbitrary accuracy [42]. Some restrictions apply; the size of the point set is allowed to be arbitrarily large and the accuracy is computed over any compact subset of $\mathbb{R}^d$ rather than the entire space.

**Definition 4.1.** *Universal Kernel* [42]: Let $S(\mathcal{X})$ denote the space of bounded and continuous functions on a compact domain $\mathcal{X}$. Then a kernel $\mathcal{K}(x, y)$ is universal if it is continuous and induces a reproducing kernel Hilbert space that is dense in $S(\mathcal{X})$.

Applied to our problem, a universal kernel is one which can approximate any well-behaved function over compact subsets of $\mathbb{R}^d$ using a linear combination of kernels. The theory of universal kernels allows for a convergence-style proof for weighted LSH kernel sums. We begin by showing that the $p$-stable LSH functions are universal kernels. We defer proofs to the appendix.

**Lemma 4.2.** *The L2 LSH kernel [19] induced by the p-stable LSH function [20] is shift-invariant and universal.*

**Theorem 4.3.** *Given a continuous and bounded function $g(q)$ and any $\epsilon > 0$, there exists a set of coefficients $\{\alpha_n\}$, set of points $\{x_n\}$ and an integer $N$ such that*

$$f_N(q) = \sum_{n=1}^{N} \alpha_N \mathcal{K}(x_n, q) \qquad \|f_N(q) - g(q)\|_{\mathcal{X}} \le \epsilon$$

*where $\mathcal{K}(x_n, q)$ is the L2 LSH kernel and $\mathcal{X}$ is any compact subset of $\mathbb{R}^d$.*

## 4.2 RIDDLE for Weighted LSH Kernel Sums

The next step of our proof is to construct a RIDDLE capable of approximating $f_N(q)$ from Theorem 4.3. We consider a RIDDLE model with $S \in \mathbb{R}^{L \times R}$ constructed from a set of points $\{x_n\}$ with

coefficient $\{\alpha_n\}$ in the following way. For every data point $x \in \{x_n\}$, we calculate the indices of $x$ using a collection of L2 LSH functions $h_l(x)$ for $l = 1, \ldots, L$. Then, we increment the value of $S$ at location $(l, h_l(x)$ with the weight $\alpha$. To query the model, we calculate the index of the query $q$ using the same LSH functions and retrieve the values $S[l, h_l(q)]$ for $l = 1, \ldots, L$, which we aggregate and return. It should be noted that recent algorithms from the sketching literature propose similar constructions [19, 18], but with the weights constrained to $\alpha_n \in \{+1, -1\}$.

One can prove $S$ yields a sharp unbiased estimator for the weighted kernel sum (Proofs in appendix).

**Theorem 4.4** (RIDDLE Estimator). *Given a dataset $\mathcal{D}$ of weighted samples $\{(\alpha_{x_i}, x_i)\}$, let $h(x)$ be an LSH function drawn from an LSH family with collision probability $\mathcal{K}(\cdot, \cdot)$. Let $S$ be a 2d parameter array constructed using $h(x)$. For any query $q$,*

$$\mathbb{E}(S[h(q)]) = \sum_i^{|\mathcal{D}|} \alpha_{x_i} \mathcal{K}(x_i, q), \operatorname{var}\left(S[h(q)]\right) \leq \left( \sum_i^{|\mathcal{D}|} \alpha_{x_i} \sqrt{\mathcal{K}(x_i, q)} \right)^2$$

The key insight is that each row of $S$ is an unbiased estimator for weighted sums of LSH kernels. By finding the central tendency of the $L$ rows, we can obtain a sharp concentration around $f_N(q)$. Although we often use the average to do the aggregation in practice, we analyze the median-of-means estimator [3, 14], as it allows us to prove an exponential concentration of the estimate around the weighted KDE. Note that while this simplifies the proof, it does not affect the convergence to $f_N(q)$ as $L \to \infty$. We combine the variance bound from Theorem 4.4 with the median-of-means guarantee to obtain a relationship between the RIDDLE parameters $S$, weighted kernel values, and estimation error.

**Theorem 4.5** (Weighted-KDE Estimation Error). *Let $Z(q)$ be the median-of-means estimate constructed using the $L$ unbiased estimators. Then with probability $1 - \delta$,*

$$|Z(q) - f_K(q)| \leq \epsilon \qquad \epsilon = 6 \frac{\tilde{f}_K(q)}{\sqrt{L}} \sqrt{\log 1/\delta}$$

*where $f_K(q)$ and $\tilde{f}_K(q)$ are the weighted KDE with kernels $\mathcal{K}(x, q)$ and $\sqrt{\mathcal{K}(x, q)}$, respectively.*

The universal approximation property of our model follows by observing that the error $\epsilon$ in Theorem 4.5 goes to zero as $N \to \infty$ and $\epsilon \to 0$ in Theorem 4.3 as $L \to \infty$.

## 5 Experiments

In this section, we first evaluate RIDDLE on practical dynamic distribution learning task in Section 5.1. Further, since we prove RIDDLE as a universal function approximator, we provide an extensive study on the expressiveness and efficiency of RIDDLE in Section 5.2. Code is available at https://github.com/lzcemma/RIDDLE.

### 5.1 RIDDLE for Learning under Dynamic Definition

Here, we investigate whether RIDDLE can adapt to new distributions while retaining good performance on the original distributions.

#### 5.1.1 Datasets

We consider four datasets that reflect dynamic distributions. Every dataset is organized into two parts: the original dataset $D^o$ and the update dataset $D^u$. $D^o$ and $D^u$ have significant semantic differences and do not share any overlapping training data.

**MNIST Binary:** We randomly partitioned digits into two groups (e.g. $4, 1, 7, 5, 3$ and $9, 0, 8, 6, 2$) and train the model to predict the group assignment of an image. For the original distribution, we randomly choose 3 digits from the first group and 3 digits from the second group such as $D^o$ only consists images of $4, 1, 7, 9, 0, 8$. The update dataset consists of images of the remaining digits.

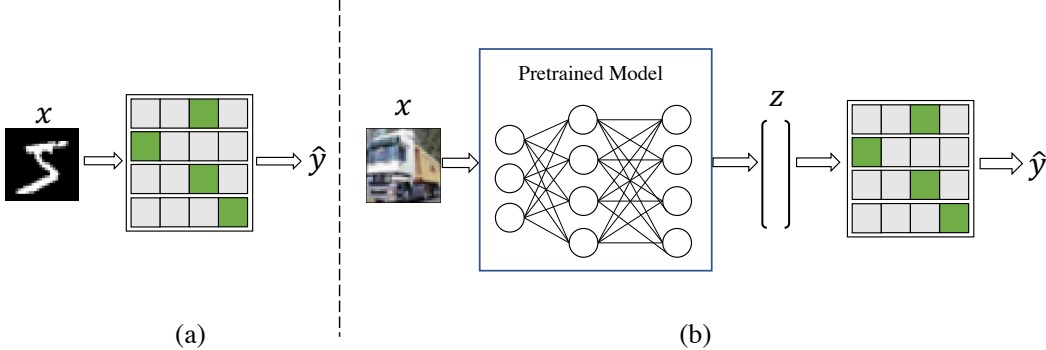

(a)                                          (b)

Figure 2: RIDDLE can be used as a stand-alone model, as shown in (a). We use this setting for experiments on MNIST-binary and all experiments studying expressiveness and efficiency. RIDDLE can also be combined with existing neural networks, as shown in (b). We replace the classification layer with RIDDLE for experiments on CIFAR10, ImageNet, and News

**CIFAR10:** The task is to predict whether the object in the image is capable of carrying things (e.g. car and horse can carry things, while a frog cannot). The original dataset $D^o$ contains images of 6 out of 10 objects that can and cannot carry things, and the update dataset $D^u$ contains the remaining 4 objects.

**ImageNet:** The task is to predict whether a person is allergic to the object in the image. We select the image classes overlapping with national allergen directctory [1], and assign images to the allergen and non-allergen classes based on the common allergen group in national allergen directory [1]. The original dataset $D^o$ contains a subset of allergens and non-allergens, while the update dataset $D^u$ contains the remaining objects. This task was built to mimic the the seasonal addition and sensitization of new allergens.

**News:** The original task for the News dataset [43] is to predict the news topics given a caption and short description. We re-task the dataset to predict whether a person would be interested in a new article by assigning some topics as interesting and the rest as not-interesing. The original dataset $D^o$ contains news from a subset of interesting and non-interesting topics, while the update dataset $D^u$ contains the rest. We organize this dataset to reflect changes in a user's interests.

### 5.1.2 Baselines

We compare RIDDLE with two updating strategies on different architectures.

**Update:** We train a neural network to full convergence on the original dataset $D^o$. Then, for updating, we train models only on the update dataset $D^u$, where the initialization is the parameters obtained on $D^o$. This approach does not require data storage nor training computation over the original dataset.

**Elastic Weight Consolidation(EWC):** [31] was proposed to mitigate catastrophic forgetting by selectively slowing down learning on the weights important for previous tasks using Fisher information.

### 5.1.3 Experiment Details

For tasks where it is reasonable (CIFAR10, ImageNet and News), we exploit recent advances in large scale pretrained models and replace the final layer with RIDDLE. Pretrained models are usually trained on a diverse corpus of unlabeled data and are likely to provide good featurization across distributions. [47] also observe that pretrained models are more resistant to forgetting.

### 5.1.4 Main Results

Table A.1 summarizes the test accuracy of all models after training on the original dataset and after the update. After training on the original dataset, we observe that all models achieve roughly the

|  |  | Trained on original dataset | | Updated on update dataset | |
|---|---|---|---|---|---|
| Dataset | Method | Original | Update | Original | Update |
| MNIST Binary | Update-MLP | 99.2 | 59.1 | 69.1 | 98.9 |
| | EWC-MLP | 99.2 | 59.8 | 90.9 | 96.7 |
| | RIDDLE | 99.2 | 60.2 | **98.3** | 98.0 |
| CIFAR10 | Update-Resnet18 | 93.3 | 69.4 | 63.22 | 88.9 |
| | Update-ViT | 93.3 | 72.6 | 74.4 | 92.7 |
| | EWC-Resnet | 92.0 | 71.8 | 75.9 | 83.9 |
| | EWC-ViT | 91.3 | 75.5 | 84.5 | 90.5 |
| | RIDDLE | 93.0 | 75.7 | **89.0** | 90.9 |
| ImageNet | Update-Resnet18 | 76.0 | 45.2 | 46.53 | 62.5 |
| | Update-ViT | 74.3 | 45.3 | 55.6 | 71.5 |
| | EWC-Resnet | 74.0 | 46.6 | 52.8 | 68.2 |
| | EWC-Vit | 71.8 | 41.2 | 54.5 | 71.8 |
| | RIDDLE | 74.3 | 45.8 | **67.1** | 70.5 |
| News | Update-RoBERTa | 96.1 | 54.8 | 74.0 | 85.7 |
| | EWC-RoBERTa | 96.5 | 45.5 | 86.1 | 87.5 |
| | RIDDLE | 96.6 | 54.6 | **89.7** | 86.4 |

Table 1: This table summarizes the accuracy after training on original dataset, and after training on update dataset. The RIDDLE obtains comparable accuracy on original test set and update test set after trained on corresponding train set. The RIDDLE's accuracy is significantly higher than all baselines on original test set after updating.

same accuracy on original test set. Accuracy on update test dataset is much worse, which is expected since the update distribution is significantly different from the original distribution.

We are interested in the accuracy on original dataset after updating the model on update dataset. We observe a 20% to 30 % accuracy drop on the original accuracy with naive updating. Models such as Resnet18 and ViT display catastrophic forgetting behavior, eventually performing no better than random. EWC is effective in prevent forgetting to some extend, reducing the degradation to around 10% to 20%. The RIDDLE significantly outperforms both baselines combined with all architectures across datasets up to in terms of original test set accuracy. Indeed, the RIDDLE keeps the accuracy degradation below 7 %. For example, the accuracy drop is only 0.9% on Mnist-Binary. On the updated test set, the RIDDLE achieves competitive accuracy (Table A.1), indicating that it is learning the properties of both distributions.

Note that ViT outperforms Resnet18 on both CIFAR10 and ImageNet, which confirms our intuition that pretrained models are inherently more robust to distribution changes. By switching the final classification layer to the RIDDLE, we increase the accuracy by a large margin. The experiment results strongly support our hypothesis that the RIDDLE retains the ability to adapt to new distributions while being less vulnerable than established methods to knowledge loss.

### 5.1.5 Ablation on Update Rule

There are two main drivers of performance when training RIDDLE over dynamically evolving definitions: the hash function and the parameter-dependently learning rate.

Without a learning rate that depends on parameter importance, the test accuracy on the original MNIST test set after updating is 94.9%, which is 26 % higher than Update-MLP, 4 % higher than EWC-MLP. Using the dynamic learning rate, RIDDLE achieves 98%. There could be smarter or simpler update functions based on the parameter count, which we leave for future work.

|  | MLP | Global Learning Rate | $\eta(c) \propto \frac{c}{c \log c + 1}$ |
|---|---|---|---|
| Test accuracy on original | 69.1 | 94.9 | 98.3 |

Table 2: This table displays the effect of the partition-dependent learning rate. "Global Learning Rate" means that the learning rate is not related to parameter counts.

| Dataset | LightGBM | Random Forest | XGBoost | NN1 | NN2 | RIDDLE |
|---|---|---|---|---|---|---|
| Susy | 0.7903 | 0.7880 | 0.7926 | 0.8019 | 0.8024 | 0.7924 |
| HAR | 0.8532 | 0.8387 | 0.8532 | 0.9027 | 0.9123 | 0.9001 |
| Covtype | 0.9565 | 0.9932 | 0.9874 | 0.9836 | 0.9936 | 0.9853 |
| Connect4 | 0.8171 | 0.8335 | 0.8171 | 0.8558 | 0.8657 | 0.8586 |
| Fashion MNIST | 0.8848 | 0.8777 | 0.8848 | 0.8852 | 0.8889 | 0.8852 |
| Boston Housing | 1.77 | 2.05 | 1.62 | 2.57 | 2.16 | 1.74 |

Table 3: This table summarizes the accuracy comparison. NN1 and NN2 denote two different deep learning models. RIDDLE achieves higher accuracy than at least half of the baselines on all datasets.

| Dataset | Method | Accuracy | Memory(MB) | Flops (M) | Inference Time (µs) |
|---|---|---|---|---|---|
| MNIST | NN | 0.9815 | 3.59 | 0.90 | 164.177 |
| | RS | 0.9784 | 1.36(1.6X) | 0.15(6.1X) | 42.003 (3.9X) |
| Fashion MNIST | NN | 0.8852 | 2.59 | 0.65 | 131.385 |
| | RS | 0.8852 | 1.92(1.6X) | 0.13(5X) | 39.228 (3.3X) |
| HAR | NN | 0.9027 | 2.11 | 0.53 | 166.584 |
| | RS | 0.9001 | 1.18(1.8X) | 0.18 (2.9X) | 46.567 (3.6X) |
| Boston Housing | NN | 2.16 | 1.96 | 0.51 | 61.338 |
| | RS | 1.74 | 0.53(3.7X) | 0.03 (17X) | 21.862 (2.8X) |

Table 4: This table summarizes the efficiency comparison between the RIDDLE and a neural network (NN) with similar accuracy. The RIDDLE reduces memory by up to 3.7x, FLOPs by up to 17x and inference time by up to 3.9x.

## 5.2 Study on Expressiveness and Efficiency

In this section, we investigate whether RIDDLE is competitive among popular machine learning models in terms of expressiveness and efficiency. Our goal is to show that RIDDLE provides a good model representation for a variety of tasks. We curated a diverse set of datasets from UCI and Kaggle. These datasets cover a wide range of tasks and domains and are widely used as benchmarks in other works [64, 22]. All experiments are conducted on a machine with 96 24-core/2-thread/2-socket processors (Intel Xeon(R) Gold 5220R 2.20GHz) and 8 Nvidia V100 32GB.

In Table 5.2, we observe that RIDDLE outperforms popular linear methods and is competitive compared to deep learning methods. NN1 and NN2 denote two different neural network architectures(Details in Appendix). Both the architecture and evaluation metric is included in Table A.2. We also investigate the efficiency and memory requirement. This is important, as it shows that RIDDLE does not outperform other methods due to an increased parameter size. Table 5.2 shows that RIDDLE attains the same accuracy as baseline methods with a smaller memory footprint, fewer FLOPs and a faster inference speed. Specifically, we use up to 3.7X less memory, use 17X less FLOPs, and achieve 3.9x faster inference speed. The memory improvements likely arise from the meaningful parameter groupings, while the computational improvements are because it only uses addition, subtraction and array lookup operations. This experiment confirms that even though the RIDDLE allocates different parameters to different distributions, we do not require more parameters than existing methods to achieve robustness or accuracy. One limitation now is that memory grows linear with number of class, which can be address with smart hashing mechanism.

## 6 Conclusion

In this work, we propose an expressive and efficient model RIDDLE for learning under dynamic distribution. Our setup concerns common scenarios where model has to be updated given distribution changes. The model is structured to be invulnerable to forgetting because gradients updates are naturally restricted to only input relevant parameters. Further, RIDDLE can be used for efficient weighted kernel density estimation.

### Acknowledgments

This work was supported by National Science Foundation SHF-2211815, BIGDATA-1838177, ONR DURIP Grant, and grants from Adobe, Intel, Total, and VMware.

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
