# OpenReview forum: "Retaining Knowledge for Learning with Dynamic Definition"
_NeurIPS.cc/2022/Conference — NeurIPS 2022 Accept_

### Official Review · Reviewer_X2yQ · 2022-07-10

**Rating:** 6
**Confidence:** 3
**Soundness:** 3 good
**Presentation:** 2 fair
**Contribution:** 3 good

**Summary:**

This paper formally defines the Learning with Dynamic Definition (LDD) problem and proposes an efficient solution, RIDDLE, to LDD. RIDDLE exploits locality-sensitive hash functions and works by assigning different sets of parameters to distinct inputs. The authors prove that RIDDLE is a universal function approximater. Theoretical analysis together with experiments on vision and NLP tasks demonstrate that RIDDLE can effectively alleviate catastrophic forgetting while achieving comparable performance with standard baseline models.

**Questions:**


1. In RIDDLE "similar" input data to use similar parameters in $S$.  This information is captured by the Counters $C$. It would be interesting to illustrate the $C$ matrix to see the pattern for different tasks. The pattern can help to show, for example, how much the update data differ from the original ones.
2. How would different hash functions affect RIDDLE?
3. What is the similarity function corresponding to the LSH function in use in RIDDLE?
4. It would be desirable to see how RIDDLE performs on normal classification tasks?
5. It would be desirable to include an ablation study to see, for example, how do $R$ and $L$ affect RIDDLE? Does the performance critically depend on the use of the dynamic learning rate $\eta(c)$?
6.  According to my understanding, RIDDLE can be dynamically extended, is this correct? For example, given a trained RIDDLE, one can increase $L$ and $R$ by smartly choosing a new hash function.



**Limitations:**

The paper briefly mentions a limitation of memory usage of RIDDLE (line 300). As far as I am concerned, the limitations also include
* a lack of ablation study.
* no experiment on standard classification tasks.
* no discussion about the similarity function corresponding to the LSH function used in RIDDLE.

**Strengths And Weaknesses:**

Strengths:
1. The central topic, i.e., alleviating catastrophic forgetting, is a crucial problem that must be resolved for the realization of Artificial General Intelligence (AGI).
2. The proposed RIDDLE model is novel, efficient, and elegant. In addition, RIDDLE may be viewed as a self-contained module that can be used together with other SOTA models.
3. Both theoretical and empirical studies are included to demonstrate the effectiveness of the proposed model.
4. The paper is generally well-written and easy to follow.

Weaknesses:
1. Missing discussion about the similarity function associated with the LSH functions in use, i.e., sparse projection-based LSH functions. Is K-L divergence the similarity function?
2. Experiments only include deliberately designed tasks, e.g., predicting whether a person is allergic to the object in the image on the ImageNet dataset, rather than the standard classification task.
3. Missing ablation study, for example, how do $L$ and $R$ affect the RIDDLE model?
4. Some symbols are confusing, for example: between lines 157 and 158, what does $\bar{c}$ mean in the formula for dynamic learning rate? and symbol $f_{S,h}(x_i)$ in Algorithm 1 is not explicitly defined.
5. The description of experiment settings needs to be more clear. For example, according to the code, the pretrained models, e.g., ViT, are frozen during training. However, this is not mentioned in the main text; and It is not clear whether all the experiments are conducted using the dynamic learning rate.
6. Several references to tables should be corrected, for example, Table 5.1 should be Table 1 (lines 264 and 275); and in line 291 it refers to Table 4 without mentioning that this table is in the supplementary material.

---

> ### Author Response · Authors · 2022-08-02
> **Reply to Review by Reviewer X2yQ**
>
> **Q1: In RIDDLE "similar" input data to use similar parameters in S. This information is captured by the Counters C. It would be interesting to illustrate the C matrix to see the pattern for different tasks. The pattern can help to show, for example, how much the update data differ from the original ones.**
>
> Thanks for the suggestions. $C$ on  $D^o$ versus $C$ on $D^u$ could be a good visualization over distribution shift. As a toy example, we plot $C_o$ and $C_u$ on Mnist_binary with $R=32$. We randomly sample 3 rows as it is easier to visualize. For each comparison, top row is from $C_o$, bottom is from $C_u$.  We can observe the density difference. Please see Appendix Figure 4 since image is not allowed in the response.
>
> **Q2: How would different hash functions affect RIDDLE?**
>
> All RIDDLE requires is that the LSH function takes the form of a positive semi-definite kernel, which includes MinHash for Jaccard similarity, p-stable LSH for Euclidean, SimHash for cosine similarity. Different LSH functions result in different partitioning of the space. This determines the types of shift to which RIDDLE is robust to. For example, because SimHash distinguishes inputs based on their angles, it will be robust to shifts along the surface of the unit sphere but not along the radius of the sphere. We refer the reader to relevant literature mentioned in Related Work for a detailed discussion on different LSH functions and their properties.
>
> **Q3: What is the similarity function corresponding to the LSH function in use in RIDDLE?**
> We use SimHash in our experiment as angular differences tend to be relevant for the input embedding from the last hidden layer.
>
> **Q3: It would be desirable to see how RIDDLE performs on normal classification tasks?**
> Please see Table 2. We curated a diverse set of datasets from UCI and Kaggle. These datasets cover a wide range of tasks and domains and are widely used as benchmarks in other works. We compare it with popular linear methods and deep learning models. This comparison is used to show RIDDLE as a general function approximater.
>
> **Q4: It would be desirable to include an ablation study to see, for example, how do R and L affect RIDDLE? Does the performance critically depend on the use of the dynamic learning rate $\eta(c)$?**
>
> We include an ablation study on $R$ and $L$ in Appendix Table 6. Since RIDDLE associates different parameters with different input, dynamic learning rate is more of a back up step. From the following table regarding accuracy on MNIST-binary,  RIDDLE still preserves most performance on original distribution, even with static learning rate.
> ||MLP |   Static learning rate  |
> | :---| :---| :---       |
> |Accuracy on Orignal Testset|  69.1/ 98.9   | 94.9 / 93.3  |
>
> **Q5: According to my understanding, RIDDLE can be dynamically extended, is this correct? For example, given a trained RIDDLE, one can increase L and  R by smartly choosing a new hash function.**
>
> This is not something that we explored in the paper, but yes. One can append new rows, maybe even rows with different hash functions.  It is possible to use different LSH functions for different rows such that RIDDLE is sensitive to multiple similarities. Also, there is no constraint to make each row the same length L.

---

> > ### Comment · Reviewer_X2yQ · 2022-08-10
> > **Thank you**
> >
> > Thanks for the detailed reply, and my concerns have been appropriately addressed.
> >
> > Considering these responses and additional experiments from the replies to other reviewers, I would like to increase my score to 6.
> >
> > At last, for the new results in your response to Q4, can you also show the performance with the static learning rate on the update dataset after updating? and, if possible, also include results on more datasets to comprehensively show how does $\eta(c)$ affect RIDDLE?

---

> > > ### Author Response · Authors · 2022-08-10
> > > **Reply to Reviewer X2yQ**
> > >
> > > Thanks for the support. We are glad our response addresses your concerns.
> > >
> > > We updated the previous response to include the accuracy on update dataset. In each cell, left number is test accuracy on original test set, right number is test accuracy on update test set. We will definitely include a more comprehensively ablation on $\eta(c)$ investigating different form of $\eta(c)$ on all datasets.

---

### Official Review · Reviewer_gwhJ · 2022-07-11

**Rating:** 6
**Confidence:** 3
**Soundness:** 2 fair
**Presentation:** 2 fair
**Contribution:** 3 good

**Summary:**

- This paper proposes RIDDLE, a model with novel parameters grouping strategy, to tackle the catastrophic forgetting problem. In this paper, the catastrophic forgetting problem is formulated as the data distribution deviations between the original and the new tasks. RIDDLE helps solve this problem by preserving the parameters that are mostly responsible for the original tasks. To be specific, RIDDLE leverages the local sensitive hash (LSH) function to map inputs to various indexes of a trainable matrix; its corresponding values are then averaged as the output of the model. Therefore, for different data distributions, LSH maps them to various indexes and utilizes the parameters from the different parts of the trainable array. Catastrophic forgetting is then prevented when the parameters responsible for original tasks are preserved. This paper theoretically shows RIDDLE can represent a large set of functions, and experimentally demonstrate how it mitigates catastrophic forgetting.

- Contributions: (a) This paper proposes a novel parameters grouping strategy. (b) This paper demonstrates the proposed framework can be used for various functions. (c) This paper shows RIDDLE is effective in multiple experiments.

**Questions:**

Based on the previous potential weakness, the questions are summarized as follows:

- The parameters updating rule:
(1) When \alpha is changing, would the model performance vary largely on original and new tasks? And would we need to search \alpha for different tasks, or it is robust to most tasks?
(2) Is this the only effective learning rate updating rule? The current updating rule seems complicated and less intuitive, and there can be some easier updating rules. For example, a simple updating rule can be using a pre-defined counting threshold $c_0$, where the parameters with frequency $c$ greater than $c_0$ will not be updated. How about the performance of those easier updating rules?

- Comparisons with other methods (e.g. EWC from Overcoming catastrophic forgetting in neural networks, and CPG from Compacting, Picking and Growing for Unforgetting Continual Learning) to prevent catastrophic forgetting: Does RIDDLE demonstrate advantages over other meta-learning methods (in terms of original /new task accuracy, efficiency, etc.)?

- Why only the last layer is replaced by RIDDLE? Specifically, what would be the performance and efficiency when the intermediate layers are also replaced by RIDDLE?

**Limitations:**

Yes. The authors have pointed out the potential limitation, and I agree with the authors that this work poses minimal negative social impact.

**Strengths And Weaknesses:**

Strengths:
- The idea of using different groups of parameters to tackle catastrophic forgetting is novel and reasonable.
- The authors demonstrate RIDDLE can be used for multiple functions and multiple tasks. This means RIDDLE can potentially be a general way to solve catastrophic forgetting for many tasks in various scenarios, which could have a significant impact.


Weakness:
- Lack of discussion on how to determine the responsible parameters. Between lines 157 and 158, the authors discuss that frequently used parameters in original tasks should be responsible for the tasks and should not be updated, and the authors propose a soft learning rate to prevent these parameters from overly updating. The proposed updating rule seems complex and is influenced by hyperparameter \alpha. How does the updating rule influence the performance in the experiment sections?

- Lack of comparisons with other methods to prevent catastrophic forgetting. There are also other methods that effectively prevent catastrophic forgetting, while the authors do not compare with them. Instead, the authors just compare with the pre-trained models. Does RIDDLE demonstrate advantages over other methods that prevent forgetting (in terms of original /new task accuracy, efficiency, etc.)?
A comparison with other methods (e.g. EWC from Overcoming catastrophic forgetting in neural networks, and CPG from Compacting, Picking and Growing for Unforgetting Continual Learning) could make this work more solid.

- Lack of reasoning on replacing the last layer with RIDDLE. In the experiments, the last layer of models is replaced by RIDDLE framework. Why only the last layer is chosen? For example, RIDDLE can also be applied in the intermediate part of the model to connect two concatenate layers.

---

> ### Author Response · Authors · 2022-08-02
> **Reply to Review by Reviewer gwhJ**
>
> Thanks for your encouragement in our novelty and significance! We appreciate your suggestions on our experiments, which have helped us improve the paper.
>
> Q1: The parameters updating rule: (1) When $\alpha$ is changing, would the model performance vary largely on original and new tasks? And would we need to search $\alpha$ for different tasks, or it is robust to most tasks? (2) Is this the only effective learning rate updating rule? The current updating rule seems complicated and less intuitive, and there can be some easier updating rules.
>
> We agree that partition dependent learning rate rules could take many reasonable functions, as long as the learning rate is inversely proportional to the counts.  In our experiment, we actually set $\alpha=0$ because $\bar{c}$ provides a good relative indicator for less used parameters. The simplest form could be $\eta(c) \propto \alpha / c$, and seems to provide reasonable performance on original test set after updating from the following table on MNIST-binary.  At the extreme case with static learning rate, RIDDLE still preserves reasonable performance on original distribution because of parameter seperation. There could be smarter/easier updating rules and we will leave it as future work.
>
> | |MLP |   Static learning rate   | $\eta(c) \propto \alpha / c, \alpha=5$    | $\eta(c) \propto \alpha / c, \alpha=10$ |
> | :---| :---| :---        |    :----:   |          ---: |
> |Accuracy on Orignal Testset|  69.1   | 94.9    | 94.7 | 93.3 |
>
> **Q2:  Does RIDDLE demonstrate advantages over other methods that prevent forgetting (in terms of original /new task accuracy, efficiency, etc.)? A comparison with other methods (e.g. EWC from Overcoming catastrophic forgetting in neural networks, and CPG from Compacting, Picking and Growing for Unforgetting Continual Learning) could make this work more solid.**
>
> Thank you for suggesting additional baselines. We have include the additional result in Appendix (Table 4 and Table 5). Given the time limit, we have performed an experimental comparison with Elastic Weight Consolidation and summarized the results in the following. All numbers are test accuracy. We apply EWC on different network architectures, e.g. EWC-Resnet18 denotes for EWC applied to Resnet18. In each cell, left number is test accuracy on original test set, right number is test accuracy on update test set. EWC is effective in improving the accuracy of the original dataset, however, we can still observe a large gap compared to RIDDLE. Also, we compared RIDDLE with the best case, retraining on all data from scratch in Appendix Table 4 . We observe only a small gap with retraining, especially on MNIST-binary, only 0.9% difference in accuracy.
>
>
> ||    Method    | **Trained on original dataset** | **Updated on update dataset**     |
> | :---| :---        |    :----:   |          ---: |
> |MNIST_Binary |   EWC-MLP    | 99.2  / 59.8    | 90.9/96.7   |
> |    | RIDDLE      | 99.2/60.2      | **98.3**/98.0 |
> |CIFAR10 |   EWC-Resnet18    | 92.0  / 71.8    | 75.9/83.9 |
> |    | EWC-Vit      | 91.3/75.5     | 84.5/90.5 |
> |    | RIDDLE      | 93.0/75.7      | **89.0**/90.9 |
> |ImageNet |   EWC-Resnet18    | 74.0  / 46.6    | 52.8/68.2 |
> |    | EWC-Vit      | 71.8/41.2     | 56.5/71.8 |
> |    | RIDDLE      | 74.3/45.8      | **67.1**/70.5 |
> |News |   EWC-RoBERTa    | 96.5  / 45.5    | 86.1/87,5 |
> |    | RIDDLE      | 96.6/54.6    | **89.7**/86.4 |
>
> **Q3: Lack of reasoning on replacing the last layer with RIDDLE. In the experiments, the last layer of models is replaced by RIDDLE framework. Why only the last layer is chosen? For example, RIDDLE can also be applied in the intermediate part of the model to connect two concatenate layers. Specifically, what would be the performance and efficiency when the intermediate layers are also replaced by RIDDLE?**
>
> We chose the classification layer because pre-trained models are usually trained on a diverse dataset, thus, more robust to distribution shift, however, classification layers are usually randomly initialized and tuned for the task during finetuning, which makes it less robust to distribution shift. Thus, the last layer is a good place to start.  Also, if only using RIDDLE as the last layer could prevent forgetting, applying RIDDLE to multiple layers could further improve the performance. We think this explains why RIDDLE has the least drop in original accuracy on MNIST-binary. We leave using RIDDLE in the intermediate layer for future work.

---

> > ### Comment · Reviewer_gwhJ · 2022-08-07
> > **Thank you**
> >
> > I would like to thank the authors for answering my questions! I would encourage the authors to continue to evaluate more baselines.

---

### Official Review · Reviewer_u21g · 2022-07-11

**Rating:** 8
**Confidence:** 2
**Soundness:** 4 excellent
**Presentation:** 3 good
**Contribution:** 3 good

**Summary:**

The paper tackles the problem of learning from new data without forgetting previously known knowledge. The authors term it Learning with Dynamic Definition (LDD). The proposed method is to hash the input and use the bucketing to associate it with the set of parameters used on it. The idea is that similar inputs will have similar associated parameters due to the hashing, so updates generated due to new inputs will likely not affect the irrelevant parameters much, thus preserving the knowledge. The authors prove that the model is a universal function approximator and theoretically bounds the knowledge lost. On many CV and NLP tasks, the authors find the method does better at retaining knowledge after the update process as evidenced by performance on the original dataset whilst also performing competitively on the new dataset.


**Questions:**

- have you explored any big datasets, how does the tradeoff for memory vs performance look like for bigger datasets?

**Limitations:**

- Limitations have already been properly/adequately addressed.

**Strengths And Weaknesses:**

 Strength:
- Relatively simple approach with good results.
- Comprehensive description of each section, well written.
- Results section and graphs are significantly impressive.

Weakness:
- I might be biased, but felt too reliant on LSH doing a good job?
- Are the scores or gains statistically significant across multiple runs?

---

> ### Author Response · Authors · 2022-08-02
> **Reply to Review by Reviewer u21g**
>
> We thank the reviewer for the strong support of our work!  We appreciate your generous comments on our thorough theoretical & empirical analysis and simple & exact algorithm.
>
> **Q1: Too reliant on LSH doing a good job?**
>
> We quantify the amount of information that is forgotten in Theorem 3.3, which is bounded by the kernel density estimate. LSH kernels are a well studied area with different LSH designed for various applications. We expect the loss to be small with a reasonable choice of LSH functions.
>
> **Q2: Have you explored any big datasets, how does the tradeoff for memory vs performance look like for bigger datasets?**
>
> Generally, bigger datasets don’t necessarily lead to larger RIDDLE. Though an increase of $L$ could reduce the variance and potentially lead to a higher accuracy. SUSY in Table2 has over 5,000,000 data. RIDDLE’s memory size is around 0.4 MB while a similar performance neural network(5 layer MLP) is around 0.7MB.

---

### Official Review · Reviewer_qrRt · 2022-07-13

**Rating:** 5
**Confidence:** 4
**Soundness:** 2 fair
**Presentation:** 2 fair
**Contribution:** 2 fair

**Summary:**

The authors propose a new model type, that they name RIDDLE, that they claim is a universal function approximator, and can outperform the performance of popular existing neural networks such as ViT and RoBERTa when it comes to catastrophic forgetting when learning on a new task.

RIDDLE consists of a number of hash functions applied to the input sample, which then are aggregated as a means to choose a parameter partition to be used for the data point at hand. They claim that the hash functions are selected such that more similar data distributions are clustered together.

They benchmark their models against a number of neural networks and more traditional linear models and show that their model exceeds most models both on the source dataset and the update dataset accuracy.

**Questions:**

1. How does your model compare with neural network based methods that selectively choose what parameters to update, such as elastic weight consolidation, or literally using Fischer information or the gradients wrt loss as a means of generating a mask for what weights to update? There are douzens of continual learning methods that work in similar principles and at least some should be compared with your method.
2. What is the optimizer and hyperparameters used? What about other training details like number of iterations and data augmentation strategies, as well as any regularization methods applied on the source and update dataset. The effect of these can be astounding.
3. Figure: What accuracy is shown on the y axis? Test set of original dataset? Or training set?


**Ethics Review Area:**

["I don’t know"]

**Limitations:**

I have already stated technical limitations, but have not noticed any negative societal impact potential from this work.

**Strengths And Weaknesses:**

Originality:
The idea itself seems quite novel, and puts a spin on the modern trends towards neural networks. It would be nice to see combinations of RIDDLE and existing NNs.

Quality:
The quality of the writing itself is fair, but the clarity of the descriptions/figures that try to distil the model ideas are hard to parse and interpret. I'd recommend taking the time to rewrite those sections more clearly and to make better figures. Figures directly comparing RIDDLE to known models structurally would be a nice way to help readers transfer their existing knowledge on to RIDDLE.

The experimental quality is problematic, as it is lacking both in terms of reporting means and standard deviations over a variety of runs, and clearly stating the metrics reported. From line 269 I can guess that the accuracies reported are on the test set, but it should be stated more explicitly, as given the target problem and the novelty of the model, one could easily assume that the authors also compared training speeds etc somewhere and thus training accuracies would also make sense. More importantly, since RIDDLE is essentially a continual learning method that uses its own selective update mechanism, it would only be fair to compare RIDDLE with other existing selective update methods for NNs, elastic weight consolitation, usage of Fischer information as a way to select what parameters to update etc, to name a few, as well as the more classic ProtoNets, Matching Networks. There are also methods that combine static backbones and inner loop updateable components such as Self Critique and Adapt. Without any of these comparisons it's hard to know whether the work is significant in any way.


Clarity:

Writing is overall OK but is also often clumsy in usage of acronyms such as CIFAR as Cifar, MNIST as Mnist -- more importantly: **The paper underplays the fact that RIDDLE is in fact using a large scale pretrained neural network as its backbone, using RIDDLE only in its last layers. ** I had to dig to find this, and found it at lines 243-246. Please make that more explicit, and rephrase writing that tries to make RIDDLE appear as something other than a neural network. I would be very keen to see how a fully RIDDLE'd model would perform when pretrained on the same dataset as the neural networks were that you used.

Also, there are no information on what optimizers and other such experimental hyperparameters mentioned. In my own work I have seen stark performance differences when fine tuning ViTs when the learning rate is not small enough. I suspect, based on the patterns in the figures of this paper that the learning rate was arbitrarily chosen or not received enough attention.




Significance:

The work can be significant for a continual learning context, but this is hard to tell as no comparisons were made with a lot of existing continual learning methods.

---

> ### Author Response · Authors · 2022-08-02
> **Reply to Review by Reviewer qrRt**
>
> We thank the reviewer for the constructive and detailed suggestions! We appreciate your generous comments on the novelty and significance. We have carefully thought through your concern and made following changes to our manuscript (1) We include additional figure( Figure 2) to clarify how we use RIDDLE within neural networks. (2) We include additional comparison in Appendix( Table 4, Table 5) (2) We include hyperparameters and training details in Appendix. A.2 (3) We fix small things such as acronyms to be consistent.  Please see the following to address your questions.
>
> **Q1:  How does your model compare with neural network based methods that selectively choose what parameters to update, such as elastic weight consolidation,ProtoNets, Matching Networks?**
>
> Thank you for suggesting additional baselines. Given the time limits, we have performed an experimental comparison with Elastic Weight Consolidation and summarized the results in the following table. We apply EWC on different network architectures, e.g. EWC-Resnet18 denotes for EWC applied to Resnet18. In each cell, left number is test accuracy on original test set, right number is test accuracy on update test set. We used Adam optimizer with lr 1e-4.  EWC is effective in improving the accuracy of the original dataset, however, we can still observe a large gap compared to RIDDLE. Also, we compared RIDDLE with the best case, retraining on all data from scratch in Table 4 . We observe only a small gap with retraining, especially on MNIST-binary, only 0.9% difference in accuracy.
>
>
> We would like to note that some suggested baselines such as ProtoNets and Matching Networks are designed for few-shot/zero-shot classification settings, where a classifier must generalize to new classes given only a small number of examples of each new class. However, this paper considers a totally different setting.  We do not introduce new classes but instead dynamically update the definition associated with existing classes.
>
>
> ||    Method    | **Trained on original dataset** | **Updated on update dataset**     |
> | :---| :---        |    :----:   |          ---: |
> |MNIST_Binary |   EWC-MLP    | 99.2  / 59.8    | 90.9/96.7   |
> |    | RIDDLE      | 99.2/60.2      | **98.3**/98.0 |
> |CIFAR10 |   EWC-Resnet18    | 92.0  / 71.8    | 75.9/83.9 |
> |    | EWC-Vit      | 91.3/75.5     | 84.5/90.5 |
> |    | RIDDLE      | 93.0/75.7      | **89.0**/90.9 |
> |ImageNet |   EWC-Resnet18    | 74.0  / 46.6    | 52.8/68.2 |
> |    | EWC-Vit      | 71.8/41.2     | 54.5/71.8 |
> |    | RIDDLE      | 74.3/45.8      | **67.1**/70.5 |
> |News |   EWC-RoBERTa    | 96.5  / 45.5    | 86.1/87.5 |
> |    | RIDDLE      | 96.6/54.6    | **89.7**/86.4 |
>
> **Q2:  What is the optimizer and hyperparameters used? What about other training details like number of iterations and data augmentation strategies, as well as any regularization methods applied on the source and update dataset.**
>
> Thank you for the suggestion. We have updated the paper to include training and model hyperparameters. Please refer to Appendix A.2. For baselines, we followed standard hyperparameters and tuned to our best.
>
> **Q3: Figure: What accuracy is shown on the y axis? Test set of original dataset? Or a training set?**
>
> Figure1 y axis shows the accuracy on the test set of the original dataset, as written in line 266 and caption.

---

> > ### Comment · Reviewer_qrRt · 2022-08-05
> > **Reply to Authors**
> >
> > Thank you for providing EWC as an additional baseline. Yes, as you state it seems that RIDDLE outperforms it.
> >
> > >We would like to note that some suggested baselines such as ProtoNets and Matching Networks are designed for few-shot/zero-shot >classification settings, where a classifier must generalize to new classes given only a small number of examples of each new class. >However, this paper considers a totally different setting. We do not introduce new classes but instead dynamically update the definition >associated with existing classes.
> >
> > ProtoNets and Matching Networks can be applied in continual settings as described in https://arxiv.org/abs/2004.11967, even when a class is continually overwritten.
> >
> > Thank you for taking care of the rest of my points.
> >
> > My main concern is that RIDDLE is still oversold as a replacement for neural networks when it relies on one in the backbone to do its job. Hence it feels like some overclaiming/distraction is taking place.
> >
> > Furthermore, there are methods far superior to EWC, say SCA (Self Critique and Adapt), which have not been compared to provide further comparisons with your method, so that more robust conclusions can be made.
> >
> > I will increase my rating, but in order to accept the paper, the overclaiming must be toned down, and RIDDLE clearly explained as a plug-and-play method on top of a learned representation (similar to how say ProtoNets use euclidean distance or MAML uses gradient-based meta-learning). Furthermore, additional comparisons with recent, and state-of-the-art continual learning methods is vital.

---

> > > ### Author Response · Authors · 2022-08-09
> > > **Reply to Additional Review by Reviewer qrRt**
> > >
> > > We appreciate your suggestions to make our paper more solid.
> > >
> > > **Q1: ProtoNets and Matching Networks can be applied in continual settings as described in https://arxiv.org/abs/2004.11967, even when a class is continually overwritten. Furthermore, there are methods far superior to EWC, say SCA (Self Critique and Adapt), which have not been compared to provide further comparisons with your method, so that more robust conclusions can be made.**
> > >
> > > Thank you for the pointer. We have added few shot continue learning in Section 2 Related Work.
> > >
> > > Besides, based on your suggestion on further baselines, we were able to run MAML as additional baselines. (SCA requires more hypterparameters tuning and code adaptation, which we couldn’t finish given the time constraint.) We used AdamW optimizer with lr 1e-3. We can see RIDDLE outperform MAML with a significant margin.
> > >
> > > ||    Method    | **Trained on original dataset** | **Updated on update dataset**     |
> > > | :---| :---        |    :----:   |          ---: |
> > > |MNIST_Binary |    MAML-MLP    | 99.2/59.4   |  90.4/90.5 |
> > > |    | RIDDLE      | 99.2/60.2      | **98.3**/98.0 |
> > > |CIFAR10 |    MAML-Resnet18      | 93.0/72.5     | 80.8/84.0 |
> > > |    | RIDDLE      | 93.0/75.7      | **89.0**/90.9 |
> > >
> > > **Q2: My main concern is that RIDDLE is still oversold as a replacement for neural networks.**
> > >
> > > Thank you for your comments regarding the positioning of the paper. We consider RIDDLE to be similar in representation capability to a one or two layer neural network. RIDDLE can solve simple tasks, such as MNIST, as a stand-alone model. For more complex tasks, RIDDLE needs to be combined with state of the art learned representations.
> > > While we did not intend to oversell RIDDLE as a replacement for deep neural networks, upon a second read through we agree that several claims can be toned down. We have made changes to  Section 1, Section 3.1 and Figure 2 to reflect the viewpoint from the previous paragraph.

---

### Comment · Area_Chair_RamJ · 2022-08-06
**Please reply to rebuttals**

Dear Reviewers,

Thanks for your work reviewing this paper. There are only a few days left for discussing with the authors.

Please read the authors' rebuttals and **explain why you decided to keep your score as is or why you updated it.** (not just click on the "acknowledge" button).  It is very frustrating for authors to be completely ignored.

Hence, we urge you to read and reply ASAP.

Thanks again,

AC

---

### Meta-Review · Area_Chair_RamJ · 2022-08-26

**Recommendation:** Accept
**Confidence:** Certain

**Metareview:**

This paper focuses on the problem of learning from new data without forgetting the knowledge from the previously learned tasks. The forgetting happens due to the parameter changes when training on new data. The author proposes RIDDLE, a model with novel parameters grouping strategy, to effectively preserve the parameters that are most responsible for the original tasks. The key idea is to leverage the local sensitive hash function to hash the input and use the bucketing to associate the input with the corresponding set of parameters. The authors conduct both theoretical and empirical analyses on the proposed method, demonstrating that RIDDLE effectively improves the performance on the original tasks, thus showing RIDDLE retains the learned knowledge. All reviewers recognize the novelty and significance of the proposed method. During the discussion, the authors also successfully addressed the reviewers' concerns on the performance comparisons with other baselines, hyperparameter settings, choice of hash functions, etc. Based on the reviews and thorough discussions, we recommend the acceptance of the paper.


**Award:**

No

---

### Decision · Program_Chairs · 2022-09-14

Accept